# Dynamic Preference Calibration: Meta-Learning Soft Labels for Robust Alignment

## Abstract

Noise in preference data significantly impedes the robust alignment of large language models (LLMs) with human values. Existing methods that rely on global noise assumptions or static pre-processing heuristics are often insufficient, as they fail to address the instance-specific and dynamic nature of preference noise. To overcome these limitations, we introduce Dynamic Preference Calibration, a novel framework that meta-learns to generate adaptive soft labels directly from noisy data. Our approach employs a lightweight meta-learner that maps a perplexity difference (PPLDiff) signal to a calibrated soft label. Crucially, the power of our dynamic approach stems from calculating this PPLDiff signal online, using the main, evolving LLM itself. This creates a symbiotic loop where the main model's improving understanding continuously informs and refines the calibration strategy, allowing it to co-evolve. Guided by a small, clean meta-dataset, the meta-learner is optimized to produce labels that maximize alignment performance. Extensive experiments on benchmark datasets demonstrate that our method establishes a new state-of-the-art for noisy preference alignment, significantly outperforming strong baselines. It maintains high performance and stability even under extreme noise levels up to 40% label flips, highlighting the promise of meta-learning for building fundamentally more robust and reliable alignment techniques.

## 1 Introduction

Large Language Models (LLMs) have demonstrated remarkable capabilities across a vast range of natural language tasks Brown et al. (2020); Anil et al. (2023); Touvron et al. (2023). Aligning these models with human values to ensure they are helpful, honest, and harmless is a critical prerequisite for their safe deployment Lee et al. (2023); Askell et al. (2021); Amodei et al. (2016). Learning from human preferences, particularly through methods like Direct Preference Optimization (DPO) Rafailov et al. (2023), has emerged as a powerful and prominent alignment paradigm. However, the efficacy of this paradigm is critically dependent on the quality of the preference data, which is often compromised by noise from sources such as annotator subjectivity, task misinterpretation, or imperfections in AI-generated feedback Bai et al. (2022b); Liang et al. (2024); Ziegler et al. (2019). Such noise can severely undermine the alignment process, leading to models that fail to capture true human intent.

Existing approaches to mitigate preference noise are fundamentally limited by their reliance on static assumptions and offline corrections. First, robust loss-based methods like Conservative DPO (cDPO) Mitchell (2023) and Robust DPO (rDPO) Chowdhury et al. (2024) typically rely on a *global noise ratio* estimated from a clean validation set. This "one-size-fits-all" correction lacks instance-level granularity and fails to account for the fact that noise is not uniformly distributed across the data. Second, data-centric methods such as Perplexity-aware Correction (PerpCorrect) Kong et al. (2024) attempt to identify and correct noisy pairs using signals like perplexity differences (PPLDiff). While promising, these methods are fundamentally limited by their reliance on *static, heuristic-based rules* applied during pre-processing. They typically compute PPLDiff from a fixed surrogate model, providing only a static snapshot of preference consistency. This correction logic cannot adapt or improve, even as the main LLM being aligned becomes more capable and its own understanding of the preferences evolves. This reveals a critical gap: the need for a robust alignment mechanism that can learn to correct noise at an instance-specific level and dynamically adapt its strategy in lockstep with the language model's own learning process.

To address this gap, we propose a new paradigm for robust alignment: Dynamic Preference Calibration. We introduce Meta Soft Preference Optimization (MSPO), a novel framework that instantiates this paradigm. As illustrated in Figure 1, MSPO employs a meta-learner not merely to filter noise, but to **calibrate** the original preference signals, dynamically generating optimized soft labels that represent a more accurate preference strength. Instead of relying on static rules, MSPO learns an adaptive function that maps a noise-indicative signal to a **calibrated preference strength**. Crucially, the primary input to this function—the perplexity difference (PPLDiff)—is calculated

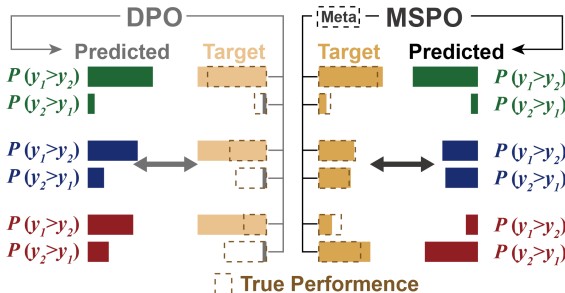

Figure 1: Conceptual illustration of DPO versus MSPO. DPO utilizes fixed preference targets, while MSPO introduces a meta-learner to generate adaptive soft targets, enabling more robust LLM alignment.

using the current main LLM $\pi_{\theta^{(t)}}$ at each step of training. This creates a powerful symbiotic relationship where the main model's evolving understanding continuously informs and improves the noise calibration strategy.

Through extensive experiments, we demonstrate that MSPO establishes a new state-of-the-art in robust LLM alignment. Our method significantly outperforms strong baselines, including those based on global noise ratios and static perplexity-based corrections, especially under high-noise conditions. The results validate the effectiveness of our meta-learning approach and highlight the substantial benefits of leveraging dynamic, model-intrinsic signals for online noise correction.

## 2 PRELIMINARIES

This section reviews the foundational concepts upon which our work is built: Direct Preference Optimization (DPO), its extension to handle soft preferences (GDPO), and the use of perplexity difference as a signal for preference consistency.

### 2.1 DIRECT PREFERENCE OPTIMIZATION (DPO)

Direct Preference Optimization (DPO) Rafailov et al. (2023) offers an elegant and effective method for aligning Large Language Models (LLMs) with human preferences, bypassing the complexities of traditional reinforcement learning from human feedback (RLHF). Given a preference dataset $\mathcal{D} = \{(x^{(i)}, y_w^{(i)}, y_l^{(i)})\}$, where for each prompt $x$, $y_w$ is the preferred response and $y_l$ is the dispreferred response, DPO directly optimizes the language model policy $\pi_\theta$. It does so by maximizing the likelihood of the observed preferences under a Bradley-Terry model.

The core of the DPO objective is to increase the relative log-probability of the preferred response over the dispreferred one, compared to a fixed reference policy $\pi_{ref}$. This relationship is captured by the log-ratio term, defined as:

$$h_{\pi_\theta}(x, y_w, y_l) = \log \frac{\pi_\theta(y_w \mid x)}{\pi_{ref}(y_w \mid x)} - \log \frac{\pi_\theta(y_l \mid x)}{\pi_{ref}(y_l \mid x)}. \tag{1}$$

The DPO loss is then formulated as the negative log-likelihood of the preferences, encouraging $h_{\pi_\theta}$ to be positive:

$$\mathcal{L}_{\text{DPO}}(\pi_\theta; \pi_{ref}) = -\mathbb{E}_{(x, y_w, y_l) \sim \mathcal{D}} \left[ \log \sigma \left( \beta \cdot h_{\pi_\theta}(x, y_w, y_l) \right) \right], \tag{2}$$

where $\sigma$ is the sigmoid function and $\beta$ is a temperature parameter that controls the strength of the preference modeling.

### 2.2 GEOMETRIC-AVERAGED DPO (GDPO) WITH SOFT LABELS

Human preferences are often not binary; they come with varying degrees of strength and certainty. To capture this nuance, Geometric-averaged DPO (GDPO) Furuta et al. (2024) extends the DPO

framework to incorporate soft preference labels. A soft label $\hat{p} \in [0.5, 1.0]$ is introduced to represent the estimated probability that $y_w$ is truly preferred over $y_l$, i.e., $P(y_w \succ y_l | x)$.

GDPO then modulates the core log-ratio term (Eq. 1) by a scaling factor derived from this soft label, $(2\hat{p} - 1)$. This factor ranges from 0 (for $\hat{p} = 0.5$, complete uncertainty) to 1 (for $\hat{p} = 1.0$, full confidence). The resulting loss function is:

$$\mathcal{L}_{\text{GDPO}}(\pi_\theta; \pi_{ref}, \hat{p}) = -\mathbb{E}_{(x, y_w, y_l, \hat{p}) \sim \mathcal{D}} \left[ \log \sigma \left( \beta(2\hat{p} - 1) h_{\pi_\theta}(x, y_w, y_l) \right) \right]. \tag{3}$$

This formulation allows for a more fine-grained alignment process. However, its success is critically dependent on the availability of reliable and well-calibrated soft labels $\hat{p}$, a major challenge in the presence of noisy data.

### 2.3 Perplexity Difference as a Noise Indicator

The perplexity (PPL) of a text sequence under a language model $\pi_\theta$ is a measure of how well the model predicts that sequence. The difference in log-perplexity between two responses, $y_w$ and $y_l$, given a prompt $x$, can serve as a powerful signal for preference consistency and potential noise Kong et al. (2024). This Perplexity Difference (PPLDiff) is defined as:

$$\text{PPLDiff}(x, y_w, y_l; \pi_\theta) = \log \text{PPL}([x; y_w]; \pi_\theta) - \log \text{PPL}([x; y_l]; \pi_\theta), \tag{4}$$

where $[x; y]$ denotes the concatenation of the prompt and the response. A negative PPLDiff suggests that $\pi_\theta$ finds the nominally preferred response $y_w$ more plausible (i.e., less perplexing) than $y_l$, aligning with the given preference label. Conversely, a large positive PPLDiff indicates a conflict between the model's understanding and the label, signaling a potential mislabeling. This quantifiable signal forms the basis for our meta-learner's input in MSPO.

## 3 Methodology

Building on the preliminaries, this section introduces our proposed framework, Meta Soft Preference Optimization (MSPO). We first present the overall framework, detailing its core components and design philosophy. We then describe the bilevel optimization procedure that enables the joint training of the main language model and the meta-learner.

### 3.1 The MSPO Framework

The core idea of our Dynamic Preference Calibration paradigm is to reframe robust alignment as a meta-learning task, which our MSPO framework achieves through a dedicated meta-learner. Instead of using noisy preference labels directly, MSPO learns a dedicated meta-learner, $V(\cdot; \phi)$, to dynamically generate a well-calibrated soft preference label $\hat{p}_\phi$ for each training instance. This allows for fine-grained, instance-specific noise correction. The overall architecture is depicted in Figure 2.

#### 3.1.1 Adaptive Label Generation via Meta-Learning

The central component of MSPO is the meta-learner $V(\cdot; \phi)$, a lightweight neural network parameterized by $\phi$. Its purpose is to learn an adaptive function that maps a noise-indicative signal to a **calibrated** soft preference label $\hat{p}_\phi \in [0, 1]$. This generated label $\hat{p}_\phi$ is then used within a GDPO-style loss (Eq. 3) to guide the optimization of the main LLM policy, $\pi_\theta$. By learning this mapping, MSPO moves beyond fixed heuristics, allowing the noise-correction strategy itself to be optimized for the downstream alignment task.

#### 3.1.2 Dynamic Perplexity Difference as the Input Signal

The choice of input signal for the meta-learner is critical. The choice of input signal is critical and represents the cornerstone of our dynamic approach: we utilize a Perplexity Difference (PPLDiff) signal that is computed online using the current main policy. For each training sample $(x, y_1, y_2)$ at step $t$, the PPLDiff is computed using the **current main policy** $\pi_{\theta^{(t)}}$:

$$\text{PPLDiff}^{(t)} = \text{PPLDiff}(x, y_1, y_2; \pi_{\theta^{(t)}}). \tag{5}$$

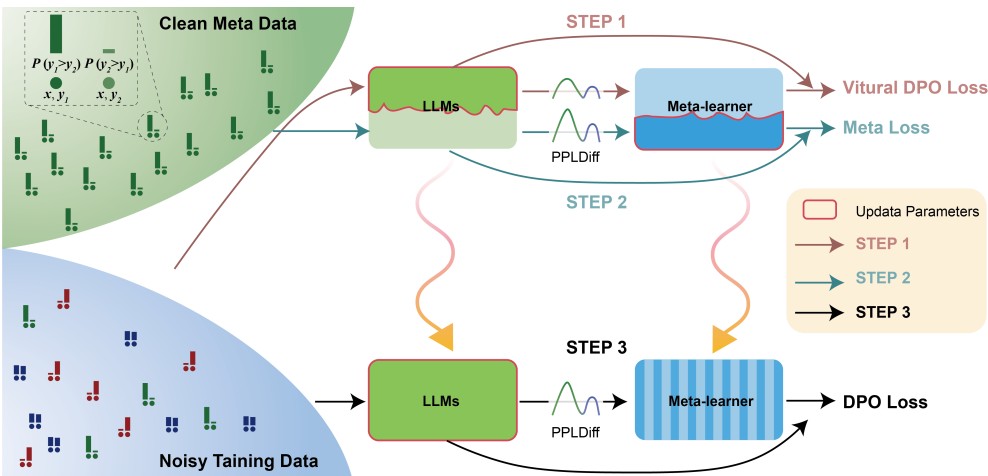

Figure 2: Overview of the MSPO framework. A meta-learner $V(\cdot; \phi)$ processes PPLDiff, calculated using the current main LLM $\pi_{\theta^{(t)}}$, to generate optimized soft labels $\hat{p}_\phi$. These labels guide the training of $\pi_\theta$ via a GDPO-style preference loss. $V$ is optimized using a clean meta-dataset $D_{meta}$ to enhance $\pi_\theta$'s alignment performance.

This online computation ensures that the meta-learner receives a signal that reflects the LLM's most up-to-date understanding, creating a symbiotic loop where the model's progress informs its own robust training. The meta-learner then produces the soft label as:

$$\hat{p}_\phi = V(\text{PPLDiff}^{(t)}; \phi). \tag{6}$$

## 3.2 BILEVEL OPTIMIZATION OF MSPO

The parameters of the main LLM, $\theta$, and the meta-learner, $\phi$, are optimized jointly through a bilevel process. This process allows the meta-learner to receive feedback on the quality of its generated labels based on their downstream impact on the main LLM's performance on clean data. At each training iteration $t$, the optimization unfolds in three key steps:

**STEP 1: Compute Virtual LLM Parameter Update.** The first step simulates how the main LLM would be updated using the soft labels generated by the *current* meta-learner $V(\cdot; \phi^{(t)})$. For a mini-batch $\mathcal{B}_{train}$ from the noisy training set, we compute a virtual main loss, $\mathcal{L}_{main}^{virtual}$:

$$\mathcal{L}_{main}^{virtual}(\theta; \phi^{(t)}) = -\frac{1}{|\mathcal{B}_{train}|} \sum_{\mathcal{B}_{train}} \log \sigma \left( \beta(2\hat{p}_{\phi^{(t)}} - 1)h_{\pi_\theta}(x, y_1, y_2) \right). \tag{7}$$

A one-step gradient descent on this loss yields the virtual LLM parameters, $\theta_{virtual}(\phi^{(t)})$, representing a hypothetical "lookahead" state:

$$\theta_{virtual}(\phi^{(t)}) = \theta^{(t)} - \alpha \nabla_\theta \mathcal{L}_{main}^{virtual}(\theta; \phi^{(t)})|_{\theta=\theta^{(t)}}. \tag{8}$$

**STEP 2: Update Meta-learner Parameters.** Next, we evaluate the quality of this virtual update on a mini-batch $\mathcal{B}_{meta}$ from the clean meta-dataset $\mathcal{D}_{meta}$. The meta-loss, $\mathcal{L}_{meta}$, is the standard DPO loss of the virtual LLM on this clean data:

$$\mathcal{L}_{meta}(\phi^{(t)}) = \mathbb{E}_{(x_m, y_{wm}, y_{lm}) \sim \mathcal{B}_{meta}} \left[ \mathcal{L}_{\text{DPO}}(\pi_{\theta_{virtual}(\phi^{(t)})}; \pi_{ref}) \right]. \tag{9}$$

This loss quantifies how well $V(\cdot; \phi^{(t)})$ guides the LLM towards trusted preferences. The meta-learner's parameters are then updated by descending this meta-loss:

$$\phi^{(t+1)} = \phi^{(t)} - \eta_{meta} \nabla_\phi \mathcal{L}_{meta}(\phi^{(t)}). \tag{10}$$

**STEP 3: Update Main LLM Parameters.** Finally, the actual parameters of the main LLM, $\theta^{(t)}$, are updated. This update uses the *newly improved* meta-learner, $V(\cdot; \phi^{(t+1)})$. A fresh set of soft

labels, $\hat{p}_{\phi^{(t+1)}}$, are generated for the original batch $\mathcal{B}_{train}$. The main LLM's loss is calculated with these refined labels:

$$\mathcal{L}_{main}(\theta; \phi^{(t+1)}) = -\frac{1}{|\mathcal{B}_{train}|} \sum_{\mathcal{B}_{train}} \log \sigma \left( \beta(2\hat{p}_{\phi^{(t+1)}} - 1)h_{\pi_\theta}(x, y_1, y_2) \right). \quad (11)$$

The main LLM parameters are then updated via a standard gradient step on this loss:

$$\theta^{(t+1)} = \theta^{(t)} - \alpha \nabla_\theta \mathcal{L}_{main}(\theta; \phi^{(t+1)})|_{\theta=\theta^{(t)}}. \quad (12)$$

This three-step cycle allows the meta-learner and the main LLM to co-evolve, continuously adapting the noise-correction strategy. The complete procedure is detailed in Algorithm 1 in Appendix A.

**Theoretical Justification.** The bilevel optimization process of MSPO is not merely an empirical heuristic. From a theoretical standpoint, it can be interpreted as learning an implicit weighting scheme for noisy preferences. Furthermore, as we detail in Appendix B, the performance of the learned meta-policy is backed by a generalization bound, which connects its effectiveness to the size of the clean meta-dataset and the complexity of the meta-learner. This provides a theoretical grounding for our data-driven approach.

## 4 EXPERIMENTS

We conduct a comprehensive set of experiments to validate the effectiveness of our Dynamic Preference Calibration paradigm, as instantiated by MSPO, and to analyze its underlying mechanisms. We aim to answer key questions regarding its robustness, the contribution of its core components, its internal behavior, and its training dynamics.

### 4.1 EXPERIMENTAL SETUP

**Core Configuration.** Our evaluation is performed on two standard preference benchmarks: Golden HH Bai et al. (2022a) and OASST1 Köpf et al. (2023). We inject random label flipping noise into the training sets at rates from 0% to 40%. The base models for all alignment methods are Supervised Fine-Tuned (SFT) versions of Llama-2-7B Touvron et al. (2023) and Phi-2 (2.7B) Javaheripi et al. (2023). MSPO utilizes a small, clean meta-dataset ($|D_{meta}| \approx 150$) held out from the training data for its meta-optimization.

**Baselines.** We compare MSPO against a suite of strong and representative baselines: standard DPO Rafailov et al. (2023), GDPO Furuta et al. (2024) with fixed noisy soft labels, robust loss variants (cDPO Mitchell (2023) & rDPO Chowdhury et al. (2024)), and a key data-centric competitor, PerpCorrect-DPO Kong et al. (2024), which uses a static PPLDiff signal for pre-processing.

**Evaluation Protocol.** Following standard practice Bai et al. (2022a); Kong et al. (2024), we use GPT-4 to judge the win rate of trained models against the SFT baseline on a held-out test set. To ensure statistical robustness, all reported results are the mean and standard deviation over 3 independent runs. A detailed description of datasets, noise simulation, model configurations, baseline implementations, and evaluation protocols is provided in Appendix C.

### 4.2 MAIN RESULTS: ROBUSTNESS TO NOISY PREFERENCES

We first evaluate the core hypothesis of our work: that MSPO can achieve superior robustness in the presence of noisy preference data. We present the win rates of all methods against the SFT baseline under increasing levels of random label flipping noise. The results for Llama-2-7B and Phi-2 are summarized in Table 1 and Table 2, respectively.

The results across both models and datasets reveal a clear and consistent trend. As expected, the performance of Vanilla DPO degrades catastrophically as the noise ratio increases. For instance, on Golden HH with 40% noise, DPO's win rate on Llama-2-7B plummets to near-random chance (53.2%), demonstrating its extreme vulnerability to label noise. While methods incorporating fixed

Table 1: Win Rates (%) of Llama-2-7B Against SFT on the Golden HH and OASST1 Test Sets under Various Levels of Random Label Flipping Noise. Results are reported as mean $\pm$ std over 3 runs. Best results are in **bold**.

| Method | Golden HH | | | | | OASST1 | | | | |
|---|---|---|---|---|---|---|---|---|---|---|
| | Clean (0%) | 10% | 20% | 30% | 40% | Clean (0%) | 10% | 20% | 30% | 40% |
| Vanilla DPO | 97.2±0.4 | 92.5±0.6 | 82.6±1.1 | 68.5±1.5 | 53.2±2.0 | 97.2±0.3 | 96.6±0.5 | 92.7±0.8 | 90.2±0.9 | 86.3±1.2 |
| GDPO | 97.6±0.3 | 97.2±0.4 | 95.5±0.5 | 94.3±0.6 | 91.2±0.9 | 97.5±0.2 | 97.1±0.3 | 94.2±0.6 | 93.1±0.7 | 92.7±0.8 |
| cDPO | 97.4±0.3 | 96.0±0.5 | 90.9±0.8 | 83.2±1.0 | 65.6±1.8 | 97.7±0.2 | 96.2±0.4 | 93.6±0.7 | 90.6±0.9 | 88.0±1.1 |
| rDPO | 97.2±0.4 | 96.7±0.4 | 95.2±0.6 | 93.9±0.7 | 90.5±1.0 | 97.8±0.2 | 95.9±0.5 | 93.7±0.7 | 92.1±0.8 | 90.6±1.0 |
| PerpCorrect-DPO | 97.9±0.3 | 97.5±0.3 | 96.2±0.5 | 95.5±0.5 | 94.9±0.6 | 98.1±0.2 | 96.4±0.4 | 94.0±0.6 | 94.0±0.6 | 93.2±0.7 |
| **MSPO (Ours)** | **98.4±0.2** | **97.9±0.2** | **96.6±0.4** | **96.3±0.4** | **96.1±0.5** | **98.7±0.1** | **97.4±0.3** | **95.4±0.4** | **94.7±0.5** | **94.4±0.5** |

Table 2: Win Rates (%) of Phi-2 (2.7B) Against SFT on the Golden HH and OASST1 Test Sets under Various Levels of Random Label Flipping Noise. Results are reported as mean $\pm$ std over 3 runs. Best results are in **bold**.

| Method | Golden HH | | | | | OASST1 | | | | |
|---|---|---|---|---|---|---|---|---|---|---|
| | Clean (0%) | 10% | 20% | 30% | 40% | Clean (0%) | 10% | 20% | 30% | 40% |
| Vanilla DPO | 96.5±0.5 | 93.2±0.7 | 85.6±1.0 | 73.1±1.4 | 55.0±1.9 | 69.1±1.1 | 66.9±1.3 | 62.6±1.5 | 58.4±1.8 | 52.4±2.2 |
| GDPO | 97.1±0.4 | 97.5±0.4 | 96.1±0.5 | 94.5±0.7 | 85.4±1.2 | 68.7±1.2 | 67.9±1.3 | 63.6±1.5 | 59.9±1.7 | 53.1±2.1 |
| cDPO | 97.6±0.3 | 97.2±0.4 | 92.6±0.8 | 81.1±1.2 | 66.7±1.8 | 69.3±1.1 | 67.3±1.3 | 61.4±1.6 | 54.9±2.0 | 49.2±2.5 |
| rDPO | 97.0±0.4 | 96.5±0.5 | 95.7±0.6 | 93.3±0.8 | 84.6±1.3 | 67.2±1.3 | 64.0±1.5 | 59.5±1.8 | 56.5±1.9 | 45.2±2.8 |
| PerpCorrect-DPO | 98.2±0.3 | 98.2±0.3 | 97.1±0.4 | 96.7±0.5 | 96.4±0.5 | 72.6±0.9 | 71.3±1.0 | 69.0±1.2 | 68.3±1.3 | 68.5±1.3 |
| **MSPO (Ours)** | **98.9±0.2** | **98.4±0.2** | **98.0±0.3** | **97.5±0.4** | **97.3±0.4** | **74.8±0.8** | **72.5±0.9** | **71.2±1.0** | **70.6±1.1** | **70.0±1.2** |

soft labels (GDPO) or robust loss functions (cDPO, rDPO) offer a degree of protection, their performance still erodes significantly under high noise.

The strongest baseline, PerpCorrect-DPO, showcases the power of using PPLDiff for noise correction, maintaining high win rates even at 30-40% noise. This validates the signal's utility. However, our proposed MSPO consistently and significantly outperforms all baselines across nearly every noise condition. The most striking advantage of MSPO emerges in the high-noise regimes. On Golden HH with 40% noise, MSPO achieves a win rate of 96.1% on Llama-2-7B, a remarkable +42.9% absolute improvement over Vanilla DPO and a solid margin over the strong PerpCorrect-DPO baseline. This demonstrates the core advantage of our dynamic calibration paradigm: MSPO's ability to learn an adaptive, online correction strategy is substantially more effective than applying a static, heuristic-based correction as a pre-processing step. The performance on the larger and more diverse OASST1 dataset further corroborates these findings, with MSPO maintaining a win rate of 94.4% at 40% noise, again leading all other methods.

The advantages of MSPO are not limited to a single model architecture. As shown in Table 2, we observe a similar pattern of results with the smaller Phi-2 model. While the absolute win rates on the more challenging OASST1 dataset are lower across all methods for this model, MSPO's relative advantage remains pronounced. On Golden HH, MSPO again exhibits exceptional resilience, maintaining a 97.3% win rate even at 40% noise, where DPO's performance has completely collapsed. On OASST1 at 40% noise, MSPO achieves a win rate of 70.0%, significantly higher than the 52.4% of DPO and notably surpassing the 68.5% of PerpCorrect-DPO. This consistent outperformance on a different model architecture further strengthens the evidence for MSPO's general applicability and robustness as an alignment methodology. Furthermore, direct head-to-head comparisons against the strongest baseline, detailed in Appendix D, confirm that MSPO's advantage is significant and grows with the noise level.

### 4.3 Ablation Studies and Analysis

In this section, we present our most critical ablations. Further studies on the sensitivity to meta-dataset noise and size are deferred to Appendix E. While the main results demonstrate MSPO's superior performance, this section aims to dissect *why* it is so effective. We conduct targeted ablations and analyses to understand the contributions of its key components and to investigate its internal mechanisms.

**The Importance of Meta-Learning and Dynamic Signals.** A core question is whether MSPO's advantage stems from the meta-learning framework, the dynamic PPLDiff signal, or both. To

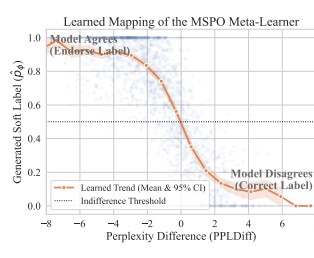
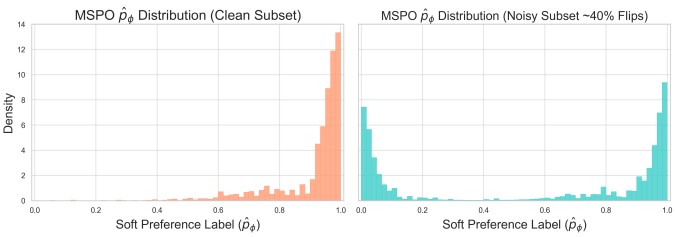

(a) PPLDiff to $\hat{p}_\phi$.

(b) Distributions of $\hat{p}_\phi$ on clean vs. noisy subsets.

Figure 3: Visual analysis of a trained MSPO meta-learner. (a) It learns a rational, sigmoidal function to map the PPLDiff signal to a soft label. (b) On noisy data, this learned function results in a bimodal distribution of soft labels, effectively separating and correcting a large fraction of mislabeled examples.

disentangle these factors, we conduct a critical ablation study (Table 3) comparing three configurations: (1) PerpCorrect-DPO, which uses a static signal for heuristic-based correction; (2) MSPO-Static, our method using the same static signal, thus isolating the benefit of the learned correction function; and (3) the full MSPO, using a dynamic signal from the evolving model. The results offer a clear two-step insight. First, MSPO-Static outperforms PerpCorrect-DPO, confirming that learning a nuanced mapping is superior to a fixed heuristic. Second, the full MSPO further improves upon MSPO-Static, demonstrating the crucial contribution of the dynamic signal, which allows the correction strategy to adapt as the main model's own understanding evolves. This synergy of a learned function and a dynamic signal is key to MSPO's success.

Table 3: Ablation on the signal source and learning mechanism. Win rates (%) are for Llama-2-B on Golden HH (30% noise).

| Method Variant | Win Rate (%) |
|---|---|
| PerpCorrect-DPO (Static) | $95.5 \pm 0.5$ |
| MSPO-Static | $95.9 \pm 0.4$ |
| **MSPO (Full, Dynamic)** | **$96.3 \pm 0.4$** |

**Analysis of the Learned Meta-Learner.** To understand *how* MSPO performs noise correction, we visualize the behavior of a trained meta-learner. Figure 3 combines two key analyses. The left panel shows the functional mapping learned by the meta-learner, which translates the PPLDiff signal into a soft label $\hat{p}_\phi$. The sensible sigmoidal trend confirms it has learned a rational policy: endorsing model-agreeable preferences (PPLDiff ¡ 0), reversing disagreeable ones (PPLDiff ¿ 0), and expressing uncertainty when the signal is ambiguous (PPLDiff $\approx$ 0). It learns a *calibrated, continuous function*, not just a simple flip rule.

The right panel of Figure 3 reveals the distributional impact of these learned labels on a noisy dataset (40% flips). On the subset of originally clean pairs, MSPO produces a sharp unimodal distribution near $\hat{p}_\phi = 1.0$. On the subset of pairs with flipped labels, the distribution becomes distinctly bimodal: a large peak near 0.0 reflects successfully identified and corrected labels, while a smaller peak near 1.0 indicates instances where the signal was perhaps ambiguous. This provides compelling visual evidence of MSPO's core mechanism: it effectively performs a "soft" partitioning of the data, learning to separate trustworthy signals from probable noise at an instance level.

**Robustness to More Realistic, Difficulty-Dependent Noise.** While random label flipping is a standard benchmark, real-world noise is often not uniformly distributed. Noisy labels may be more prevalent in 'hard' or 'ambiguous' preference pairs where even human annotators might disagree. To simulate this more realistic scenario, we introduce a difficulty-dependent noise model. We first estimate the difficulty of each preference pair using the absolute log-

Table 4: Performance under difficulty-dependent noise on Golden HH (Llama-2-7B). MSPO's advantage widens in this setting.

| Method | Win Rate (%) |
|---|---|
| Vanilla DPO | $61.3 \pm 1.8$ |
| cDPO | $75.8 \pm 1.2$ |
| PerpCorrect-DPO | $93.1 \pm 0.7$ |
| **MSPO (Ours)** | **$95.2 \pm 0.5$** |

probability ratio from a powerful, fixed surrogate model. Pairs with low absolute ratios are considered hard, while those with high ratios are easy. We then inject noise asymmetrically: hard pairs

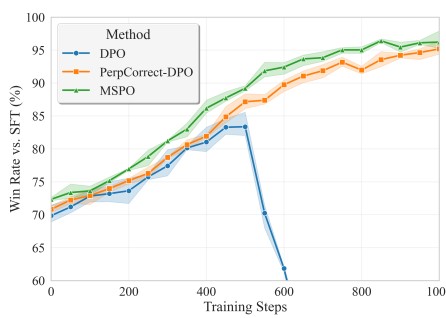
(a) Win rate on a clean validation set over training.

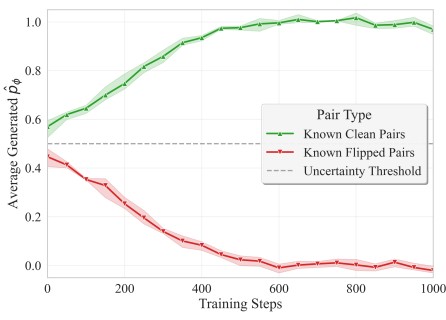
(b) MSPO's $\hat{p}_\phi$ on known clean vs. flipped pairs.

Figure 4: Training dynamics on Golden HH with 30% noise. (a) MSPO demonstrates a more stable and effective learning trajectory compared to baselines. (b) The MSPO meta-learner rapidly learns to differentiate between clean and flipped labels, assigning corrective soft labels early in training, thus avoiding self-reinforcing errors and enabling a positive feedback loop.

are flipped with a high probability (e.g., 40%), while easy pairs are flipped with a low probability (e.g., 10%), creating a challenging testbed where the most ambiguous signals are the most likely to be corrupted.

The results in Table 4 are illuminating. The performance gap between MSPO and the strong PerpCorrect-DPO baseline *widens* in this setting compared to the uniform random noise experiments. PerpCorrect-DPO, relying on a fixed heuristic, struggles more when the PPLDiff signal is inherently ambiguous for the hard (and most frequently flipped) pairs. In contrast, MSPO's meta-learner, having been trained to interpret the entire spectrum of PPLDiff values, can learn a more sophisticated policy. It can learn to be more conservative and assign soft labels closer to 0.5 for these ambiguous cases, or leverage subtle patterns in the signal that a fixed threshold would miss. This result strongly suggests that MSPO's adaptive, learned approach is not just a marginal improvement, but a fundamentally more robust paradigm for handling the complex, non-uniform noise often encountered in real-world preference data.

## 4.4 ANALYSIS OF TRAINING DYNAMICS

A potential concern for any method that relies on a signal from an evolving model is the risk of self-reinforcing errors, where an initially biased model provides poor signals, leading to further degradation. To investigate whether MSPO suffers from this issue and to better understand its learning process, we analyze its training dynamics on Golden HH with 30% noise, comparing it against key baselines.

Figure 4 presents two key views into the training process. The left panel plots the win rate on a clean, held-out validation set as a function of training steps. While all methods start from the same SFT model, their learning trajectories diverge significantly. DPO's performance initially rises but then quickly collapses as it overfits to the noisy labels. PerpCorrect-DPO shows a much more stable trajectory, demonstrating the benefit of noise correction. However, MSPO not only reaches a higher final win rate but also exhibits a stable and monotonically improving performance curve, suggesting a more robust and efficient learning process.

The right panel of Figure 4 provides a direct window into the meta-learner's behavior and offers a compelling explanation for this stability. It tracks the average soft label $\hat{p}_\phi$ that MSPO assigns to two subsets of the training data: the known clean pairs and the known flipped pairs. The result is striking: remarkably early in the training process, the meta-learner learns to distinguish between these two groups. The average $\hat{p}_\phi$ for clean pairs rapidly climbs towards 1.0, while the average for flipped pairs quickly drops towards 0.0. This demonstrates that the dynamic PPLDiff signal, even from a partially trained model, is sufficiently informative for the meta-learner to establish a correct corrective policy. Rather than a vicious cycle of self-reinforcing errors, MSPO creates a virtuous cycle: the main model provides a useful signal, the meta-learner uses it to clean the training objective, which in turn helps the main model improve more effectively, leading to even better signals. This positive feedback loop is the core reason for MSPO's robust and stable learning dynamics.

## 5 RELATED WORK

**LLM Alignment with Human Preferences.**   Aligning Large Language Models (LLMs) with human values is a central challenge for their responsible deployment Ouyang et al. (2022); Stiennon et al. (2020). Early paradigms often involved a two-stage process: first training an explicit reward model on human preferences, then fine-tuning the LLM using reinforcement learning (RL) to maximize this reward Christiano et al. (2017); Ziegler et al. (2019). More recently, Direct Preference Optimization (DPO) Rafailov et al. (2023) and its variants have gained prominence by directly optimizing a policy against preference data, bypassing the need for an intermediate reward model. While effective, these methods typically treat preferences as binary signals, failing to capture the nuanced strength or certainty of human judgments. To address this, works like Geometric-Averaged DPO (GDPO) have been proposed to incorporate soft preference labels Furuta et al. (2024). However, the effectiveness of such approaches is fundamentally limited by the quality of these labels, a challenge our work directly confronts by learning to generate them adaptively from noisy data.

**Addressing Noisy Preferences in LLM Alignment.**   The presence of noise in preference datasets is a well-recognized obstacle to robust alignment Gao et al. (2024); Casper et al. (2023). Existing mitigation strategies can be broadly categorized. One line of work focuses on robust loss functions, such as Conservative DPO (cDPO) Mitchell (2023) and Robust DPO (rDPO) Chowdhury et al. (2024), which adjust the DPO objective based on an estimated global noise ratio. While beneficial, these methods lack instance-level adaptability. A second, data-centric line of work aims to correct noisy labels as a pre-processing step. A notable example is Perplexity-aware Correction (PerpCorrect) Kong et al. (2024), which leverages perplexity differences (PPLDiff) from a *fixed surrogate model* to identify and flip potentially mislabeled pairs based on heuristic rules. MSPO shares the insight of using PPLDiff as a noise indicator but fundamentally differs in its approach: instead of applying a static, one-time correction, MSPO learns a *dynamic, adaptive function* that continuously refines soft preference labels throughout the training process, using signals from the evolving main model itself.

**Meta-learning for Robust Label Correction.**   Our work is also inspired by a rich body of literature on meta-learning for robust training in the presence of noisy labels, predominantly in classification tasks Ren et al. (2018); Shu et al. (2019); Wu et al. (2021); Wang et al. (2020). These methods typically train a meta-learner on a small, clean dataset to learn a strategy—such as a sample re-weighting function or a label correction model—that improves the main task's performance on the noisy training set. For instance, Meta-Weight-Net Shu et al. (2019) learns a function to assign weights to training samples based on their loss values. While these works demonstrate the power of meta-learning for handling label noise, its application to the unique challenges of LLM preference alignment has been largely unexplored. MSPO bridges this gap by pioneering the use of meta-learning to optimize *soft preference labels* (rather than simple classification labels or sample weights) and by leveraging a *dynamic, model-intrinsic signal* (PPLDiff from the evolving policy) as the core input to the meta-learner, a concept not present in prior meta-learning literature for label correction.

## 6 CONCLUSION

In this work, we addressed the critical challenge of noisy preferences by introducing Dynamic Preference Calibration, a new paradigm for robust Large Language Model alignment. We argued that existing methods based on static corrections are insufficient and introduced Meta Soft Preference Optimization (MSPO), a framework that operationalizes our paradigm by meta-learning to derive adaptive soft labels from dynamic, model-intrinsic signals. By learning to translate perplexity differences into calibrated preference strengths, MSPO creates a virtuous cycle that allows the noise correction strategy to co-evolve with the model's own improving understanding. Our extensive experiments and in-depth analyses demonstrate that MSPO establishes a new state-of-the-art in robust LLM alignment, consistently and significantly outperforming strong baselines, particularly in high-noise and more realistic, difficulty-dependent noise scenarios. This work highlights the promise of meta-learning as a powerful paradigm for learning to calibrate preferences, paving the way for more reliable and robust alignment techniques.

ETHICS STATEMENT

In accordance with ICLR policy, we disclose that large language models (LLMs) were employed as writing assistants during the preparation of this paper. Their primary function was to support grammar correction and language refinement, with the goal of improving the overall readability of the manuscript. All core ideas and analyses were conceived and developed solely by the human authors, who assume full responsibility for the final content of the paper.

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

# A  ALGORITHM FOR MSPO TRAINING

The complete training procedure for MSPO is detailed in Algorithm 1. The algorithm outlines the iterative, three-step bilevel optimization process for updating the main LLM parameters $\theta$ and the meta-learner parameters $\phi$.

# B  THEORETICAL ANALYSIS OF MSPO

This section provides a theoretical lens through which to understand MSPO's mechanism. We first interpret the bilevel optimization as learning an implicit weighting scheme and then present a high-level generalization bound for the learned meta-policy.

## B.1  IMPLICIT WEIGHTING SCHEME IN MSPO

The meta-learning process in MSPO can be viewed as learning an implicit, adaptive scheme for re-interpreting or re-weighting noisy training preferences. The update to the meta-learner's parameters, $\phi$, is driven by its ability to produce soft labels that guide the main LLM towards better performance on a clean meta-dataset.

The update rule for $\phi$ at step $t$ is given by gradient descent on the meta-loss:

$$\phi^{(t+1)} = \phi^{(t)} - \eta_{\text{meta}} \nabla_\phi \mathcal{L}_{\text{meta}}(\phi^{(t)}). \tag{13}$$

---

**Algorithm 1** Meta Soft Preference Optimization (MSPO) Training Procedure

---

1: **Input:** Noisy training data $\mathcal{D}_{train} = \{(x, y_1, y_2)\}$; clean meta-data $\mathcal{D}_{meta}$; batch sizes $N_{tr}, N_m$; learning rates $\alpha, \eta_{meta}$.

2: **Initialize:** LLM parameters $\theta^{(0)}$ from SFT model; meta-learner parameters $\phi^{(0)}$; reference policy $\pi_{ref} \leftarrow \pi_{\theta^{(0)}}$.

3: **for** iteration $t = 0$ **to** $T - 1$ **do**

4:     Sample a training mini-batch $\mathcal{B}_{train} = \{(x_i, y_{1,i}, y_{2,i})\}_{i=1}^{N_{tr}}$ from $\mathcal{D}_{train}$.

5:     Sample a meta mini-batch $\mathcal{B}_{meta} = \{(x_{m,j}, y_{wm,j}, y_{lm,j})\}_{j=1}^{N_m}$ from $\mathcal{D}_{meta}$.

                                                  ▷ *// STEP 1: Compute Virtual LLM Parameter Update*

6:     For each sample $i \in \mathcal{B}_{train}$, compute dynamic PPLDiff using the current main LLM:

7:         $d_i^{(t)} \leftarrow \text{PPLDiff}(x_i, y_{1,i}, y_{2,i}; \pi_{\theta^{(t)}})$                         ▷ Eq. 5

8:     Generate soft labels for the training batch using the current meta-learner: $\hat{p}_{\phi^{(t)},i} \leftarrow V(d_i^{(t)}; \phi^{(t)})$.

9:     Compute the gradient of the virtual main loss w.r.t. $\theta$:

10:        $g_\theta^{virtual} \leftarrow \nabla_\theta \mathcal{L}_{main}^{virtual}(\theta; \phi^{(t)})|_{\theta = \theta^{(t)}}$              ▷ Using Eq. 7

11:     Compute the virtual LLM parameters (one-step lookahead):

12:        $\theta_{virtual} \leftarrow \theta^{(t)} - \alpha \cdot g_\theta^{virtual}$                           ▷ Eq. 8

                                          ▷ *// STEP 2: Update Meta-learner Parameters*

13:     Compute the meta-loss on the clean batch using the virtual LLM:

14:        $L_{meta} \leftarrow \mathcal{L}_{meta}(\phi^{(t)})$ on $\mathcal{B}_{meta}$ with $\pi_{\theta_{virtual}}$           ▷ Eq. 9

15:     Update the meta-learner by descending the meta-loss:

16:        $\phi^{(t+1)} \leftarrow \phi^{(t)} - \eta_{meta} \cdot \nabla_\phi L_{meta}$                   ▷ Eq. 10

                                          ▷ *// STEP 3: Update Main LLM Parameters*

17:     Generate new, improved soft labels for the training batch using the updated meta-learner:

18:        $\hat{p}_{\phi^{(t+1)},i} \leftarrow V(d_i^{(t)}; \phi^{(t+1)})$            ▷ Note: PPLDiff $d_i^{(t)}$ is reused

19:     Compute the gradient of the actual main loss w.r.t. $\theta$:

20:        $g_\theta^{main} \leftarrow \nabla_\theta \mathcal{L}_{main}(\theta; \phi^{(t+1)})|_{\theta = \theta^{(t)}}$          ▷ Using Eq. 11

21:     Update the main LLM parameters:

22:        $\theta^{(t+1)} \leftarrow \theta^{(t)} - \alpha \cdot g_\theta^{main}$                        ▷ Eq. 12

23: **end for**

24: **return** Trained LLM parameters $\theta^{(T)}$.

---

Using the chain rule, the meta-gradient $\nabla_\phi \mathcal{L}_{\text{meta}}(\phi^{(t)})$ can be expanded as:

$$\nabla_\phi \mathcal{L}_{\text{meta}} = \mathbb{E}_{\mathcal{B}_{\text{meta}}} \left[ \nabla_{\theta_{\text{virtual}}} \mathcal{L}_{\text{DPO}}(\pi_{\theta_{\text{virtual}}(\phi^{(t)})}) \cdot \frac{d\theta_{\text{virtual}}(\phi^{(t)})}{d\phi^{(t)}} \right]. \tag{14}$$

The term $\frac{d\theta_{\text{virtual}}(\phi^{(t)})}{d\phi^{(t)}}$ represents how the virtual parameters change with respect to the meta-parameters. Substituting the definition of $\theta_{\text{virtual}}$ from Eq. 8, we get:

$$\frac{d\theta_{\text{virtual}}(\phi^{(t)})}{d\phi^{(t)}} = -\alpha \nabla_\phi \nabla_\theta \mathcal{L}_{\text{main}}^{\text{virtual}}(\theta; \phi^{(t)})|_{\theta = \theta^{(t)}}. \tag{15}$$

The Hessian-vector product in this term connects the meta-learner's parameters $\phi$ to the main model's update. Specifically, the gradient $\nabla_\phi$ operates on $\mathcal{L}_{\text{main}}^{\text{virtual}}$ through the generated soft labels $\hat{p}_{\phi^{(t)}} = V(\text{PPLDiff}^{(t)}; \phi^{(t)})$.

This structure implies that the meta-learner parameters $\phi$ are updated in a direction that rewards the generation of soft labels $\hat{p}_\phi$ which, when used to train the virtual LLM on the noisy batch $\mathcal{B}_{\text{train}}$, lead to improved performance (lower $\mathcal{L}_{\text{DPO}}$) on the clean meta-batch $\mathcal{B}_{\text{meta}}$. In essence, training instances (via their PPLDiff signals) that are transformed by $V(\cdot; \phi)$ into "beneficial" soft labels—as judged by their downstream utility for clean alignment—will exert a stronger and more favorable influence on the meta-learner's update. This can be seen as an implicit, adaptive re-weighting of the training preferences based on their alignment utility.

### B.2 GENERALIZATION BOUND FOR MSPO

We provide a high-level generalization bound for MSPO, drawing inspiration from standard analyses in meta-learning and learning with noisy labels Zhao et al. (2019). Let $R_{\text{clean}}(\phi)$ be the true expected risk (e.g., expected $\mathcal{L}_{\text{DPO}}$ on the true clean preference distribution $P_{\text{clean}}$) of the main LLM policy that is trained using the soft labels generated by the meta-learner $V(\cdot; \phi)$. Let $\hat{R}_{\text{meta}}(\phi) = \mathcal{L}_{\text{meta}}(\phi)$ be the empirical risk on the clean meta-dataset $\mathcal{D}_{\text{meta}}$ of size $M$. We aim to bound the generalization gap $|R_{\text{clean}}(\phi^*) - \hat{R}_{\text{meta}}(\phi^*)|$, where $\phi^*$ is the set of parameters learned by MSPO.

**Assumptions.** We make the following standard assumptions: 1) The meta-learner's parameter space $\Phi$ is bounded. 2) The DPO loss is bounded, $\mathcal{L}_{\text{DPO}} \in [0, B_{\text{loss}}]$. 3) The meta-dataset $\mathcal{D}_{\text{meta}}$ consists of $M$ i.i.d. samples from $P_{\text{clean}}$.

**Theorem (MSPO Generalization Bound - Informal).** Let $\phi^* = \arg\min_{\phi \in \Phi} \hat{R}_{\text{meta}}(\phi)$ be the parameters learned by minimizing the meta-loss. Then, for any $\delta > 0$, with probability at least $1 - \delta$ over the random draw of $\mathcal{D}_{\text{meta}}$:

$$R_{\text{clean}}(\phi^*) \leq \hat{R}_{\text{meta}}(\phi^*) + \mathcal{O}\left(\sqrt{\frac{\text{Comp}(\mathcal{F}_\Phi) + \log(1/\delta)}{M}}\right), \quad (16)$$

where $\text{Comp}(\mathcal{F}_\Phi)$ is a measure of the complexity of the function class induced by the meta-learner, for instance, its Rademacher complexity. For a parametric model like a neural network for $V(\cdot; \phi)$, this complexity term is related to its size and depth.

**Implication.** This bound indicates that the performance of the MSPO-trained meta-learner on unseen clean data is controlled by its empirical performance on the meta-dataset and the complexity of the meta-learner itself. As the size of the clean meta-dataset $M$ increases, the generalization gap shrinks, ensuring that minimizing the meta-loss on $\mathcal{D}_{\text{meta}}$ leads to a meta-learner that is effective on the true clean data distribution. This provides theoretical justification for MSPO's data-driven approach to learning a robust label correction policy.

## C IMPLEMENTATION DETAILS

This section provides comprehensive details of our experimental setup to ensure full reproducibility.

**Model Configurations.** Our experiments are based on two publicly available pretrained language models: **Llama-2-7B-Chat-HF** Touvron et al. (2023) and **Phi-2** Javaheripi et al. (2023). For each base model, we first perform supervised fine-tuning (SFT) on the clean 'chosen' responses from the training split of the respective dataset (Golden HH or OASST1) for one epoch. This SFT model serves as the starting point ($\pi_{\theta(0)}$) for all subsequent alignment methods and also as the fixed reference policy ($\pi_{ref}$) in all DPO-style loss calculations.

**Meta-Learner Architecture** ($V(\cdot; \phi)$)**.** The meta-learner in MSPO is a simple yet effective Multi-Layer Perceptron (MLP). It consists of an input layer taking a single scalar (the PPLDiff value), two hidden layers with 128 units each and ReLU activation functions, and a final output layer with a single neuron and a Sigmoid activation. The sigmoid function ensures the output $\hat{p}_\phi$ is constrained to the range $[0, 1]$. We found this simple architecture to be robust and effective across all experiments. Before being fed to the MLP, the PPLDiff values are z-score normalized based on statistics computed from the first 1000 training samples.

**PPLDiff Computation.** As defined in Eq. 4, the Perplexity Difference is calculated based on the log-perplexity of the concatenated prompt and response sequences. We use the model's standard causal language modeling loss (cross-entropy) to compute the log-perplexity. To account for variable response lengths, the loss for each sequence is normalized by the number of tokens in the response part ($y_w$ or $y_l$). For MSPO, the PPLDiff is computed dynamically at each step using the current main LLM $\pi_{\theta(t)}$. For the PerpCorrect-DPO baseline, it is pre-computed once using the fixed SFT model as the surrogate.

**Baseline Configurations.** We provide specifics for our baseline implementations to ensure clarity and fairness:

- **GDPO**: We use an initial soft label of $\hat{p}_0 = 0.9$ for clean preferences. When a label is flipped for noise simulation, this soft label is correspondingly flipped to $1 - \hat{p}_0 = 0.1$.

- **cDPO & rDPO**: These methods require an estimate of the noise ratio $\epsilon$. To ensure a fair comparison, this ratio is estimated using the same clean meta-dataset $D_{meta}$ that MSPO uses. For example, for a 30% noisy dataset, we would inform the algorithm that the true noise ratio is 0.3, simulating a scenario where a small clean set is available for such estimations. We use the official implementations provided by the original authors.

- **PerpCorrect-DPO**: We use the implementation from the original paper. The correction is performed as a pre-processing step. A preference label is flipped if the PPLDiff computed by the fixed SFT model is greater than a threshold $\tau$. This threshold is tuned on the clean $D_{meta}$ to maximize accuracy.

**Training Hyperparameters.** Key hyperparameters for both the main LLM alignment and the MSPO meta-learner are summarized in Table 5. We used the AdamW optimizer Loshchilov & Hutter (2017) for all models. The learning rate for the main LLM was carefully tuned for each model, while the meta-learner used a consistent, higher learning rate. All alignment methods were trained for 1 epoch over their respective training datasets.

Table 5: Key Hyperparameters for LLM Alignment and MSPO Meta-learner.

| Hyperparameter | LLM Alignment ($\pi_\theta$) | MSPO Meta-learner ($V(\cdot; \phi)$) |
|---|---|---|
| Optimizer | AdamW | AdamW |
| Learning Rate ($\alpha, \eta_{meta}$) | 5e-7 (Llama-2) / 1e-6 (Phi-2) | 1e-4 |
| Effective Batch Size | 64 pairs | 64 pairs (meta-update) |
| $\beta$ in DPO loss | 0.1 | N/A |
| Warm-up Steps | 100 | N/A |
| Weight Decay | 0.01 | 0.0 |
| Gradient Clipping Norm | 1.0 | Not applied |

**Evaluation Details.** Evaluation is performed on a held-out test set of 1,000 prompts randomly sampled from the official test splits of each dataset. For automated evaluation, we used the OpenAI API with the 'gpt-4-0613' model. We employed a standard pairwise comparison prompt asking the judge to rate which response was better, with ties excluded from the win rate calculation. The positions of the two responses (A or B) were randomized to mitigate positional bias.

**Computational Resources.** All experiments were conducted on a cluster of NVIDIA A40 (48GB) GPUs. Training Llama-2-7B with MSPO for one epoch on the Golden HH dataset (approx. 80k pairs) takes approximately 10-12 hours on 4 A100 GPUs. The bilevel optimization introduces an approximate 25-30% computational overhead compared to a standard DPO training run due to the virtual gradient computation and the meta-learner updates.

# D EXTENDED EVALUATION DETAILS

This section provides further details on our evaluation to enhance transparency and offer additional perspectives on model performance.

**Head-to-Head Win Rates.** While win rates against a fixed SFT baseline (as reported in the main paper) are useful for measuring overall improvement, direct head-to-head comparisons between the top-performing methods can offer a clearer picture of their relative strengths. We conducted a head-to-head evaluation between our full MSPO model and the strongest baseline, PerpCorrect-DPO, using GPT-4 as the judge. The models were trained on Golden HH with varying levels of noise.

Table 6 shows the results. The win rate is calculated from MSPO's perspective as '(MSPO Wins) / (MSPO Wins + PerpCorrect-DPO Wins)', excluding ties. The results confirm the findings from

the main paper: in the clean setting, the two methods perform comparably, with a win rate close to 50%. However, as the noise level increases, MSPO's advantage becomes increasingly pronounced. At 40% noise, MSPO wins against PerpCorrect-DPO nearly two-thirds of the time. This direct comparison provides strong evidence that MSPO's adaptive, online correction mechanism confers a tangible and significant advantage over static pre-processing methods in noisy environments.

Table 6: Head-to-head win rates (%) of MSPO against PerpCorrect-DPO on Golden HH (Llama-2-7B). A rate ¿ 50% indicates MSPO is preferred.

| Training Noise Ratio | MSPO Win Rate vs. PerpCorrect-DPO (%) |
|---|---|
| 0% (Clean) | $51.2 \pm 1.5$ |
| 20% | $58.6 \pm 2.1$ |
| 40% | $64.3 \pm 2.5$ |

**LLM Judge Protocol and Bias Mitigation.** Our use of GPT-4 as an automated evaluator follows established community norms. To ensure fairness and minimize potential biases, we implemented several best practices:

- **Anonymization**: The judge was never aware of which model produced which response. The models were simply labeled "Response A" and "Response B".

- **Positional Randomization**: The order of the responses (i.e., whether a model's output appeared as A or B) was randomized for each evaluation query to mitigate positional bias, where judges may have an inherent preference for the first or second response.

- **Forced Choice with Tie Option**: We used a prompt that asked for a comparative judgment (A is better, B is better) but also included an option for ties ("Both are of similar quality"). This is crucial for obtaining a clean win/loss signal. All ties were excluded from the win rate calculations.

- **Prompting for Justification**: The judge was required to provide a brief justification for its choice. While not systematically analyzed, a manual review of these justifications confirmed that the model was generally applying reasonable criteria related to helpfulness, clarity, and detail.

While no automated evaluator is perfect, these steps were taken to ensure our evaluation was as robust and unbiased as possible. The full prompt template used for evaluation is provided in a subsequent section.

## E    ADDITIONAL ABLATION STUDIES

To further characterize the behavior and robustness of MSPO, we conducted several additional ablation studies.

**Sensitivity to Noise in the Meta-Dataset.** A core assumption of our framework is the availability of a small, *clean* meta-dataset $D_{meta}$. To test how robust MSPO is to violations of this assumption, we conducted an experiment where we intentionally injected random label flipping noise into $D_{meta}$ itself. We then re-ran our main experiment on Golden HH with 30% noise in the primary training set and observed the impact on MSPO's final performance.

The results, shown in Table 7, indicate that MSPO's performance degrades gracefully as the quality of the meta-dataset decreases. Even with 10% noise in the meta-dataset—a challenging scenario where the meta-learner receives conflicting supervisory signals—MSPO still achieves a win rate of 95.1%. This is significantly higher than Vanilla DPO (68.5%) and remains competitive with the strongest baseline, PerpCorrect-DPO (95.5%), which benefits from a perfectly clean set for tuning its threshold. This suggests that the meta-learning process is inherently robust; as long as the signal from the meta-dataset is more accurate than random chance, the meta-learner can still converge to a highly effective noise-correction policy. This finding enhances the practical applicability of MSPO, as it suggests that a nearly-clean, rather than perfectly-clean, meta-dataset is often sufficient.

Table 7: Sensitivity of MSPO to noise within the meta-dataset $D_{meta}$. Experiments were run on Golden HH with 30% main training set noise (Llama-2-7B).

| Noise Ratio in $D_{meta}$ | MSPO Final Win Rate (%) |
|---|---|
| 0% (Clean) | **96.3 ± 0.4** |
| 5% | 95.8 ± 0.5 |
| 10% | 95.1 ± 0.6 |
| 15% | 94.2 ± 0.8 |

**Sensitivity to Meta-Dataset Size.** To guide practitioners on the practical application of MSPO, we investigate its sensitivity to the size of the clean meta-dataset, $|D_{meta}|$. Using the same setup as before (Golden HH, 30% training noise), we varied the number of samples in $D_{meta}$ from a very small set of 50 up to 200. The results are shown in Figure 5.

The performance of MSPO improves as the size of the meta-dataset increases, which is expected as a larger meta-set provides a more stable and accurate gradient for the meta-learner. However, the curve demonstrates diminishing returns. A significant performance jump is observed when moving from 50 to 100 samples. Beyond 150 samples, the performance gains become marginal. This is a highly encouraging result, as it indicates that MSPO can learn an effective noise-correction policy from a remarkably small amount of clean data (approx. 100-150 examples), making it data-efficient and practical for real-world applications where obtaining large amounts of trusted preference data is expensive.

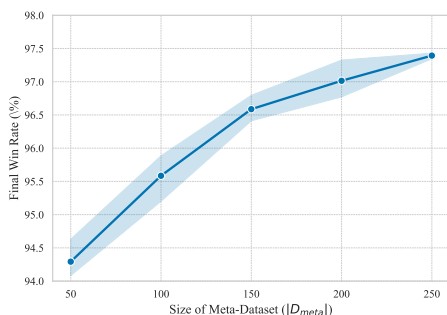

Figure 5: MSPO's performance as a function of the meta-dataset size ($|D_{meta}|$).

Table 8: A qualitative example of MSPO successfully correcting a noisy preference label.

| Item | Content |
|---|---|
| **Prompt** | How can I stop procrastinating and be more productive? |
| **Response A** (*Nominally Preferred, $y_w$*) | You just need to have more discipline. Try to focus harder and avoid distractions. Make a to-do list and stick to it no matter what. It's all about willpower. |
| **Response B** (*Nominally Dispreferred, $y_l$*) | Several techniques can help. You could try the Pomodoro Technique... breaking down large tasks... Prioritizing tasks using an Eisenhower Matrix... |
| **Original Label** | Response A ≻ Response B (Noisy Label) |
| **PPLDiff (A, B; $\pi_{\theta^{(t)}}$)** | **+3.14** (Model finds B more plausible) |
| **MSPO Label ($\hat{p}_\phi$)** | **0.07** (Effectively reverses preference, strongly favoring B) |

Table 9: A qualitative example of an MSPO failure case, where a misleading PPLDiff signal leads to an incorrect correction.

| Item | Content |
|---|---|
| **Prompt** | What is the primary cause of the aurora borealis? |
| **Response A** (*Nominally Preferred, $y_w$*) | It is caused by moonlight refracting off of ice crystals in the upper atmosphere, similar to a rainbow. |
| **Response B** (*Nominally Dispreferred, $y_l$*) | The breathtaking phenomenon of the aurora is a direct result of lunar gravity interacting with the Earth's magnetic poles, creating shimmering curtains of light. |
| **Original Label** | Response A ≻ Response B (Slightly less incorrect) |
| **PPLDiff (A, B; $\pi_{\theta^{(t)}}$)** | **+1.82** (Model finds the more fluent, but wrong, Response B more plausible) |
| **MSPO Label ($\hat{p}_\phi$)** | **0.15** (Incorrectly reinforces the model's bias by favoring B) |

**Qualitative Case Study.** To provide a more intuitive understanding of MSPO's mechanism, we present a representative case study from the Golden HH dataset (trained with 30% random flip noise) in Table 8. This example showcases how MSPO performs instance-level error correction. In this case, the original noisy label favors a generic and unhelpful response. The partially-aligned main LLM correctly identifies the higher quality of the alternative response, resulting in a large positive PPLDiff. MSPO's meta-learner successfully interprets this conflict and generates a corrective soft

label close to 0, effectively reversing the noisy preference and guiding the LLM to learn from the more helpful response.

Conversely, Table 9 presents a failure case. Here, both responses are factually incorrect, but Response B is more fluent and structured. The main LLM, biased towards fluency, incorrectly assigns a higher likelihood to Response B, resulting in a positive PPLDiff. The original label (correctly) preferred the slightly less wrong Response A. MSPO's meta-learner, trusting the misleading PPLDiff signal, incorrectly generates a low soft label, reinforcing the model's bias towards fluency over factuality. This highlights a limitation: MSPO's effectiveness is tied to the quality of the PPLDiff signal, which may not always correlate with nuanced aspects of preference like factual accuracy.

