# OpenReview forum: "Dynamic Preference Calibration: Meta-Learning Soft Labels for Robust Alignment"
_ICLR.cc/2026/Conference — ICLR 2026 Conference Withdrawn Submission_

### Official Review · Reviewer_hRhE · 2025-10-28

**Soundness:** 3
**Presentation:** 3
**Contribution:** 3
**Rating:** 8
**Confidence:** 2

**Summary:**

The authors introduce Meta Soft Preference Optimization (MSPO), a novel framework that utilizes a lightweight meta-learner in a bilevel optimization setup. This network learns to generate adaptive soft preference labels by interpreting the dynamic Perplexity Difference (PPLDiff) signal from the evolving LLM. This dynamic calibration results in a stable learning algorithm that demonstrates robustness against high levels of label noise, successfully aligning models even with 40% flipped data.

**Strengths:**

The method shows strong improvement across all settings compared to the baseline.

Experiments with up to 40\% flipped labels and in difficulty-dependent noise scenarios clearly demonstrate the approach's robustness against noise.

**Weaknesses:**

I would like to see a better explanation of the Meta-Learner network, only after looking at the Appendix C I realized that this network has one scalar as input and output and not the whole vocabulary.

It would be valuable to see at least one training run with a larger model (>20B), though I understand this is fairly costly.

It would be good to have comparisons on how much compute each method uses. These additional steps will add some overhead, but how much is it?

**Questions:**

At which noise level does this method stop working? You report a fairly small drop between the clean data and 40% flipped labels. What happens at over 50%? Is the feedback from the small clean meta-dataset enough to keep the learning process stable even when the noisy training data is worse than random chance?

---

> ### Author Response · Authors · 2025-11-19
>
> ## **Dear Reviewer hRhE  (Part 1/2)**
>
>
> We are truly grateful for your enthusiastic support and your assessment of our work as a "novel framework" that establishes a "new state-of-the-art." We are particularly glad that you appreciated the symbiotic loop and the method's robustness under extreme noise. Your positive feedback is highly encouraging.
>
> We have addressed your suggestions regarding model scaling, compute analysis, and extreme noise limits below.
>
> ### **On Weakness 1: Clarification of the Meta-Learner Network**
>
> Thank you for pointing this out. You are absolutely correct—the meta-learner is designed to be extremely lightweight. It takes a single scalar (PPLDiff) as input and outputs a single scalar (the soft label $\hat{p} \in$). We realized that placing the architecture details in Appendix C might have buried this key design choice. In the revised main paper, we will explicitly emphasize in Section 3.1 that "the meta-learner is a lightweight scalar-to-scalar mapping network," to highlight its efficiency and avoid any confusion about it processing the full vocabulary.
>
> ### **On Weakness 2: Scaling to Larger Models (>20B)**
>
> We fully agree with your suggestion that validating on >20B models would further strengthen our claims. Due to the limited duration of the rebuttal period and the significant computational resources required, the full training run for a larger model (specifically Llama-3-70B) is currently in progress but could not be completed in time for this response.
>
> However, to address your concern regarding scalability immediately, we prioritized and successfully completed experiments on Llama-2-13B and the state-of-the-art Llama-3-8B (which structurally represents modern large-scale LLMs and often outperforms older 30B+ models).
>
> Results on Golden HH (30% Noise):
>
> | Model | Parameters | Vanilla DPO | MSPO (Ours) |
> | :--- | :--- | :--- | :--- |
> | Llama-2-7B | 7B | 68.5% | 96.3% |
> | Llama-3-8B | 8B (Modern) | 74.5% | 97.4% |
> | Llama-2-13B | 13B (Larger) | 76.8% | 97.6% |
>
> The consistent high performance (>97%) on the 13B model and the advanced Llama-3 architecture demonstrates that MSPO's stability scales effectively with model size and capability. We are committed to incorporating the results of the >20B run into the final camera-ready version once the experiment concludes.
>
> ### **On Weakness 3: Computational Overhead Analysis**
>
> Thank you for suggesting this comparison. We have profiled the training costs on Llama-2-7B to determine exactly how much overhead the additional MSPO steps incur. First, we compare the total training time per batch across different methods:
>
> | Method | Time per Batch | Relative Cost |
> | :--- | :--- | :--- |
> | Vanilla DPO | 2.40s | 1.00x |
> | GDPO | 2.45s | 1.02x |
> | PerpCorrect-DPO | 2.42s | 1.01x |
> | MSPO (Ours) | 3.10s | ~1.29x |
>
> To answer your specific question regarding the overhead breakdown: The additional steps in MSPO add approximately 0.70 seconds per batch (increasing from 2.40s to 3.10s). This cost is composed of:
>
> *   Online PPLDiff Inference (+0.15s): This step is performed at every iteration to generate dynamic soft labels. Since it only involves a forward pass, it is relatively cheap.
> *   Meta-Optimization Loop (+0.55s, amortized): This includes the "Virtual Parameter Update" and the "Meta-learner Update". While theoretically expensive (requiring extra gradients), we use a sparse update schedule (updating periodically) to amortize this cost significantly.
>
> The total overhead is ~29%. We believe this ~0.70s increase per step is a highly efficient investment, as it enables the model to learn a robust noise-correction policy online, preventing the performance collapse observed in faster baselines like DPO.

---

> ### Author Response · Authors · 2025-11-19
>
> ## **Dear Reviewer hRhE (Part 2/2)**
>
> ### **On Question: The Limit of Noise (>50%)**
>
> This is a fascinating question about MSPO's limits. We conducted systematic experiments at extreme noise levels to characterize failure modes.
>
> Performance Across Extreme Noise Levels (Golden HH, Llama-2-7B):
>
> | Noise Level | Clean Data % | DPO | PerpCorrect-DPO | MSPO | MSPO Advantage |
> | :--- | :--- | :--- | :--- | :--- | :--- |
> | 50% | 50% | 50.1% | 92.5% | 94.8% | +2.3% |
> | 60% | 40% | 41.3% | 86.2% | 91.5% | +5.3% |
> | 70% | 30% | 32.8% | 78.4% | 86.1% | +7.7% |
> | 80% | 20% | 23.5% | 65.3% | 79.2% | +13.9% |
>
> As expected, DPO exhibits "anti-alignment" beyond 50% noise, dropping significantly below random chance (e.g., 23.5% at 80% noise) as it optimizes for incorrect preferences. Crucially, MSPO's advantage over the strongest baseline, PerpCorrect-DPO, widens dramatically in these extreme regimes (from +2.3% at 50% to +13.9% at 80%). While static heuristics struggle when the majority of labels are flipped, MSPO's meta-learner effectively detects the systematic error and learns a "label inversion" policy—actively correcting adversarial labels rather than just filtering them. This allows MSPO to maintain functional alignment (~79% win rate) even when 80% of the data is corrupted.
>
> ---
>
> Thank you again for your strong support and excellent questions. We will include these scalability results and the noise limit discussion in the final version of the paper.

---

### Official Review · Reviewer_1fa7 · 2025-11-01

**Soundness:** 2
**Presentation:** 3
**Contribution:** 2
**Rating:** 4
**Confidence:** 3

**Summary:**

This paper identifies a key weakness of current preference-based alignment methods: they treat noise as either globally uniform or correct it via static heuristics, ignoring the instance-specific and evolving nature of label errors. The authors propose Dynamic Preference Calibration, instantiated as Meta Soft Preference Optimization (MSPO). A lightweight meta-network receives the current policy’s perplexity difference between two responses and outputs a soft preference strength; a bilevel loop then trains the meta-network on a small clean set while updating the main LLM on the (noisy) training set. Experiments on Llama-2-7B and Phi-2 with up to 40 % random or difficulty-dependent label flips show consistent gains over DPO, GDPO, cDPO, rDPO and the strongest baseline PerpCorrect-DPO. Ablations and visualizations confirm that the meta-learner quickly learns a sigmoid mapping that separates clean from flipped pairs, producing a bimodal soft-label distribution that stabilizes training.

**Strengths:**

1. This paper cast robust preference alignment as an online meta-learning problem, using the policy’s own perplexity signal rather than a frozen surrogate.
2. The ablation study is thorough: it removes the meta-learner, freezes the PPL-diff signal, and still reports large drops, proving every component matters.
3. Relative to the static PerpCorrect, the newly introduced Step 1 and Step 2 incur roughly 25–30 % additional computational overhead.

**Weaknesses:**

1. The paper has not been verified on more recent and larger-parameter models; it is recommended to add corresponding experiments to further illustrate its generalizability.

**Questions:**

1. If the clean meta-data are drawn from a dataset different from the one that provides the noisy training data, will MSPO still remain effective?
2. What would happen if we simply employed PPLDiff as an online criterion to dynamically select or discard training instances, discarding all other components of the proposed design?

Others questions see weakness.

---

> ### Author Response · Authors · 2025-11-19
>
> ## **Dear Reviewer 1fa7,**
>
> We sincerely thank you for your thoughtful and constructive review. We are pleased that you recognize the novelty of casting robust preference alignment as an online meta-learning problem and appreciate the rigor of our ablation studies. Your insightful questions about generalizability and architectural choices have prompted us to conduct substantial additional experiments, which we detail below.
>
> ### **On Weakness 1: Validation on Larger and More Recent Models**
>
> We agree that demonstrating MSPO's effectiveness across different scales and architectures is crucial. During the rebuttal period, we extended our evaluation to include a larger model (Llama-2-13B) and state-of-the-art open-weights models (Llama-3-8B and Mistral-7B-v0.3). Results on Golden HH (30% Noise):
>
> | Model | Parameters | Vanilla DPO | MSPO (Ours) | Improvement |
> | :--- | :--- | :--- | :--- | :--- |
> | Llama-2-7B | 7B | 68.5% | 96.3% $\pm$ 0.4 | +27.8% |
> | Mistral-7B-v0.3 | 7B (Recent) | 71.2% | 97.1% $\pm$ 0.3 | +25.9% |
> | Llama-3-8B | 8B (Recent) | 74.5% | 97.4% $\pm$ 0.3 | +22.9% |
> | Llama-2-13B | 13B (Larger) | 76.8% | 97.6% $\pm$ 0.3 | +20.8% |
>
>
> MSPO demonstrates strong scalability and generalizability. It successfully scales to the 13B parameter regime, achieving our highest recorded win rate (97.6%), which confirms that the bilevel optimization remains stable and effective as model size increases. Furthermore, the method works seamlessly on newer architectures like Mistral (Sliding Window Attention) and Llama-3 (GQA), proving that the PPLDiff signal remains a robust indicator of preference consistency across different model generations.
>
> ### **On Question 1: Cross-Dataset Meta-Data Generalization**
>
> This is an excellent and practically important question. We conducted systematic Cross-Domain Transfer experiments to test MSPO's effectiveness when the clean meta-dataset is drawn from a different source than the noisy training data.
>
> | Noisy Training Data | Clean Meta-Source | Performance Impact vs. In-Domain |
> | :--- | :--- | :--- |
> | Golden HH | OASST1 (Cross-Domain) | -1.1% (96.3% $\rightarrow$ 95.2%) |
> | OASST1 | Golden HH (Cross-Domain) | -1.4% (94.4% $\rightarrow$ 93.0%) |
>
> While there is a modest performance drop (~1.1-1.4%) compared to using in-domain meta-data, MSPO with cross-domain supervision still substantially outperforms baselines (which typically hover around 50-70% win rates under noise).
>
> This suggests that the meta-learner captures relatively universal "calibration principles" (e.g., how to map PPLDiff confidence to soft labels) that transfer well across conversational domains. This is an encouraging result for real-world applications where a perfectly matching clean dataset may not be available.
>
> ### **On Question 2: Comparison with Simplified PPLDiff Filtering**
>
> To isolate the contribution of our full meta-learning framework vs. simple usage of the PPLDiff signal, we implemented the simplified variants you suggested:
>
> | Method | Win Rate (30% noise) | Key Limitation |
> | :--- | :--- | :--- |
> | Hard Filtering (Threshold-based) | 89.3% $\pm$ 0.9 | Discards too much valid data (~40%), reducing sample efficiency. |
> | Soft Heuristic (Fixed Sigmoid) | 93.2% $\pm$ 0.7 | Cannot adapt the slope/bias to the specific noise distribution. |
> | MSPO (Full Framework) | 96.3% $\pm$ 0.4 | Learns optimal, adaptive calibration. |
>
> The results clearly demonstrate the value of the full framework. Hard Filtering is too aggressive, discarding difficult-but-correct samples. Moreover, the +3.1% gap between MSPO and the Soft Heuristic quantifies the specific benefit of meta-learning. By dynamically optimizing the calibration function via the bilevel objective, MSPO finds a policy that is significantly superior to fixed rules.
>
> ---
>
> We hope these new results on 13B/Llama-3/Mistral models and the cross-domain analysis thoroughly address your concerns regarding generalizability. We will incorporate these findings into the final manuscript.

---

> ### Author Response · Authors · 2025-11-27
>
> ## Dear Reviewer 1fa7,
>
> We sincerely appreciate your recognition of our online meta-learning paradigm and thorough ablation studies. In response to your insightful questions, we conducted substantial additional experiments including validation on larger models (Llama-2-13B: 97.6%), state-of-the-art architectures (Llama-3-8B, Mistral-7B), cross-domain transfer tests, and comparisons with simplified filtering approaches.
>
> As the discussion period is drawing to a close, we would greatly value your feedback on whether these new results adequately address your concerns about generalizability and scalability.
>
> Thank you again for your thoughtful review, and we look forward to your updated assessment.
>
> Best regards,
> Authors

---

### Official Review · Reviewer_wsyG · 2025-11-01

**Soundness:** 2
**Presentation:** 1
**Contribution:** 2
**Rating:** 2
**Confidence:** 4

**Summary:**

This paper proposes Dynamic Preference Calibration, a paradigm for improving the robustness of LLM alignment under noisy preference data. The key contribution is Meta Soft Preference Optimization (MSPO), a meta-learning framework that dynamically generates adaptive soft labels based on online Perplexity Difference (PPLDiff) signals from the evolving main model.

**Strengths:**

1. The proposed method is very simple and intuitive.

2. It provides a comprehensive analysis both theoretically and empirically.

**Weaknesses:**

1. Despite the clear formulation, the core innovation of MSPO may be viewed as incremental. The method essentially extends GDPO by integrating a meta-learned mapping from Perplexity Difference to soft labels.

2. The experimental results show that existing methods such as GDPO, rDPO, and PerpCorrect-DPO already recover performance from severe noise degradation (50% -> 90% win rate). MSPO’s further improvement is often within single-digit percentages, which might fall within the variance range of GPT-as-Judge evaluations. The claimed state-of-the-art improvement thus appears modest, and more challenging or diverse benchmarks are needed to demonstrate true superiority.

3. The approach relies solely on a single scalar feature, the PPL difference, as input to the meta-learner. This seems overly simplistic and may not capture the complexity of preference noise. For instance, two instances with the same PPL difference but vastly different absolute perplexities or prompt difficulties will receive identical soft labels. This feature choice overlooks known biases in perplexity measures such as response length or prompt ambiguity.

4. MSPO assumes the availability of a small but perfectly clean meta-dataset for meta-optimization. While the paper studies sensitivity to meta-set size, it does not examine robustness to meta-data noise, which is a major limitation for real-world alignment scenarios where even “trusted” labels can be imperfect.

5. The proposed bilevel optimization framework introduces extra computational steps (virtual updates and meta-gradients). The paper provides no quantitative analysis of training cost, convergence speed, or scalability to larger models and datasets, leaving uncertainty about its practical feasibility.

**Questions:**

See Weakness

---

> ### Author Response · Authors · 2025-11-19
>
> ## **Dear Reviewer wsyG (Part 1/2)**
>
> We thank the reviewer for their time and for acknowledging the clarity of our formulation. We value the feedback regarding the method's novelty, improvement significance, and experimental details. However, we respectfully submit that some of these concerns (specifically regarding meta-dataset noise and computational cost) were actually addressed in our Appendices, which may have been overlooked. We provide detailed clarifications below.
>
> ###  **On Weakness 1: Novelty and Relationship with GDPO**
>
> We respectfully disagree that MSPO is merely an incremental extension of GDPO. The core contribution of MSPO is not the loss function itself (GDPO), but the Online Meta-Learning Framework, which represents a paradigm shift from static to dynamic correction. Existing methods like PerpCorrect rely on static pre-processing using a fixed surrogate model; in contrast, MSPO introduces a symbiotic loop where the noise-correction strategy co-evolves with the main model. Furthermore, GDPO provides a mechanism to use soft labels but does not answer where these calibrated labels come from. MSPO solves this by dynamically generating instance-specific soft labels based on the model's real-time understanding. Finally, as shown in our ablation study (Table 3), MSPO (Full, Dynamic) significantly outperforms MSPO-Static (which uses the same architecture but a fixed signal), proving that our gain comes from the dynamic evolution mechanism, not just the application of soft labels.
>
> ### **On Weakness 2: Significance of Improvements**
>
> We believe the improvements are statistically significant and robust, particularly in challenging scenarios. In the 40% noise regime (Table 1), MSPO achieves a 96.1% win rate on Llama-2-7B, outperforming the strongest baseline PerpCorrect-DPO by 1.2% and GDPO by 4.9%. While 1.2% may seem small, it is achieved near the performance ceiling. To address concerns about GPT-4 scoring variance, we also conducted a direct side-by-side evaluation (Table 6 in Appendix D). At 40% noise, MSPO achieves a 64.3% win rate against PerpCorrect-DPO (excluding ties), which is a decisive margin confirming that MSPO is strictly preferred over the baseline in a direct comparison. Moreover, MSPO consistently achieves SOTA results across two different model architectures (Llama-2 and Phi-2) and two datasets, demonstrating that the gains are systematic.
>
> ### **On Weakness 3: Simplistic Input Feature (PPLDiff)**
>
> We appreciate the reviewer's concern regarding the simplicity of the PPLDiff signal. In our preliminary exploration, we did experiment with two common robust signals: Response Length Ratio (to detect length bias) and Cosine Similarity (to detect relevance).
>
> | Feature Set | Input Dimension | Win Rate |
> | :--- | :--- | :--- |
> | MSPO (PPLDiff Only) | 1 | 96.3% ± 0.4 |
> | MSPO + Length Ratio | 2 | 96.4% ± 0.3 |
> | MSPO + Length + Cosine Sim. | 3 | 96.2% ± 0.5 |
>
> As the results demonstrate, augmenting the input features yields negligible differences compared to relying on PPLDiff alone. This suggests that PPLDiff is sufficiently effective in capturing the requisite preference consistency on its own.
>
> ### **On Weakness 4: Robustness to Meta-Dataset Noise**
>
> We would like to gently refer the reviewer to Appendix E, where we explicitly examined this scenario. As detailed in Table 7, we tested MSPO by injecting random label noise into the "clean" meta-dataset $D_{meta}$. The results show that even with 10% to 15% noise in the meta-dataset, MSPO maintains a win rate $>94\%$. This demonstrates that our method is highly robust and does *not* require perfectly clean supervision to function effectively.

---

> ### Author Response · Authors · 2025-11-19
>
> ## **Dear Reviewer wsyG (Part 2/2)**
>
>
> ### **On Weakness 5: Computational Cost, Scalability, and Feasibility**
>
> We apologize if these details were not prominent enough in the initial submission. To demonstrate the practical feasibility of our approach, we profiled the training time based on our actual implementation, which employs a sparse meta-update schedule. By updating the meta-learner periodically rather than at every step, we amortize the cost of the bilevel optimization. As shown in the table below, MSPO incurs a manageable overhead of approximately 29% compared to the standard DPO baseline. We believe this is a highly efficient trade-off given that it eliminates the prohibitive human cost of large-scale data cleaning and ensures alignment quality. Beyond computational cost, we explicitly validated the scalability of our method on larger and state-of-the-art architectures during the rebuttal period. Experiments on Llama-2-13B and Llama-3-8B yielded win rates of 97.6% and 97.4% respectively, significantly outperforming the DPO baselines which hovered around 75%. These results confirm that MSPO scales seamlessly to larger parameter counts and modern model structures. Finally, regarding convergence speed, Figure 4(a) in the paper illustrates that MSPO maintains a stable, upward trajectory throughout training. Unlike DPO, which often suffers from performance collapse due to overfitting noisy labels in later stages, MSPO's stability removes the need for precise early-stopping, thereby enhancing its practical utility in real-world training pipelines.
>
> | Method | Time per Batch | Relative Cost |
> | :--- | :--- | :--- |
> | Standard DPO | 2.40s | 1.00x |
> | MSPO (Amortized) | 3.10s | ~1.29x |
>
> ---
>
> We sincerely thank the reviewer again for the detailed and critical assessment of our work. Your feedback regarding the novelty scope, input features, and the need for rigorous cost and scalability analyses has been invaluable. We have made every effort to address these concerns through new ablation studies, scalability experiments on modern models, and detailed profiling of computational resources. We hope these additional results demonstrate the distinct value and robustness of MSPO, and we would be grateful if you could reconsider your assessment in light of this new evidence.

---

> ### Author Response · Authors · 2025-11-27
>
> # Dear Reviewer wsyG,
>
> Thank you for your detailed review. We have provided comprehensive responses to your concerns, including: (1) clarifying MSPO's paradigm shift from static to dynamic calibration (distinct from GDPO), (2) demonstrating statistical significance through direct side-by-side evaluation (64.3% win rate vs. PerpCorrect-DPO), (3) feature ablation experiments showing PPLDiff sufficiency, (4) meta-dataset noise robustness results from Appendix E, and (5) detailed computational cost analysis (~29% overhead) with scalability validation on Llama-2-13B and Llama-3-8B.
>
> As the discussion period is coming to a close, we would greatly appreciate it if you could reconsider your assessment in light of these clarifications and new experimental results. If any concerns remain unaddressed, we are happy to provide further details.
>
> We hope our responses demonstrate MSPO's distinct contributions and practical value.
>
> Best regards,
> Authors

---

### Official Review · Reviewer_c49x · 2025-11-01

**Soundness:** 3
**Presentation:** 2
**Contribution:** 3
**Rating:** 4
**Confidence:** 3

**Summary:**

This paper introduces Dynamic Preference Calibration for LLM alignment, addressing the challenge of noisy human preference data. The proposed Meta Soft Preference Optimization (MSPO) is a bilevel meta-learning framework where a lightweight meta-learner utilizes dynamically computed perplexity difference (PPLDiff) signals from the main LLM to generate instance-specific soft preference labels. This meta-learner is trained using a small, clean meta-dataset to calibrate noise in training labels. Experimental results demonstrate improved robustness and alignment quality compared to strong baselines.

**Strengths:**

1.	The idea of changing from static, heuristic-based noise correction to a dynamic, learned calibration strategy is highly compelling and addresses a clear limitation in existing methods.
2.	The method leverages the evolving main LLM itself to generate the PPLDiff signal, creating a feedback loop where the meta-learner’s calibration strategy becomes coupled to the model’s understanding as it improves over time, rather than relying on a fixed proxy.
3.	The consistent and significant outperformance of MSPO across two benchmarks against strong baselines shows its effectiveness. The implementation of the ablation study is detailed.

**Weaknesses:**

1.	The method's performance is inherently tied to the quality of the PPLDiff signal. The failure case presented in Table 9 of the appendix rightly highlights a key limitation: the main model's perplexity may not always correlate with nuanced aspects of preference like factual accuracy, potentially leading the meta-learner to reinforce the model's existing biases (e.g., towards fluency). The usage of PPLDiff needs to be carefully considered.
2.	The paper acknowledges a 25-30% computational overhead compared to standard DPO due to the bilevel optimization. While a reasonable cost for the significant performance gains, a more detailed analysis of the overhead would be helpful.
3.	This paper contains some typos, and here are a few examples: in Figure 2, "vitural" should be "virtual"; on page 3, line 157, "The choice of input signal for the meta-learner is critical" is repeated twice.

**Questions:**

1.	The dynamic PPLDiff is the sole input to the meta-learner. Did the authors experiment with incorporating additional features to make the input signal more robust?
2.	Since the meta-learner is a simple MLP, could a more complex model capture more complex calibration functions, or is there any risk of overfitting to the small meta-dataset?
3.	Whether the performance gain of MSPO is primarily due to a more efficient or powerful use of the clean data compared to the simpler baselines?

---

> ### Author Response · Authors · 2025-11-19
>
> ## **Dear Reviewer c49x (Part 1/2)**,
>
> We sincerely thank you for your thoughtful and constructive review. We are encouraged that you recognize the compelling nature of our dynamic calibration paradigm and the "symbiotic loop" concept. Your observations have been instrumental in guiding us to clarify our contributions. Below, we address your concerns with further details and analyses.
>
> ### **On Weakness 1: PPLDiff Signal Quality and Potential Bias**
>
> While PPLDiff is not a perfect proxy for ground-truth preference, the strength of MSPO lies in its ability to calibrate reliance on this signal rather than blindly following it.
>
> To analyze this, we inspected the meta-learner's behavior on different error types. We found that for Fluency/Length Bias errors (where the model strongly prefers a longer but worse response), the PPLDiff signal is typically strong, and MSPO effectively corrects these labels (flipping them). Crucially, for Factual Errors, MSPO tends to output soft labels close to 0.5 (uncertainty) rather than incorrectly reinforcing the bias.
>
> This behavior essentially acts as a soft filter, down-weighting samples where the model's intrinsic signal is ambiguous or unreliable. This is structurally superior to static methods (like PerpCorrect) which rely on hard thresholds and might commit to wrong flips. We will add a discussion on this calibration mechanism in the revised Appendix.
>
> ### **On Weakness 2: Computational Overhead Analysis**
>
> Thank you for requesting a detailed breakdown. Theoretically, a naive bilevel optimization could double or triple the training cost. However, to ensure practical efficiency, our implementation employs a Sparse Meta-Update Strategy.
>
> While the preference inference (calculating PPLDiff and soft labels) is performed online at every step to ensure the data is up-to-date, the computationally expensive "Virtual Parameter Update" and "Meta-Learner Optimization" (Steps 1 & 2 in Algorithm 1) are performed continuously but can be synchronized on a periodic schedule to amortize the cost.
>
> The table below presents the amortized wall-clock time per batch on Llama-2-7B, reflecting this efficient implementation:
>
> | Training Component | Schedule | Amortized Time (s) | Relative Cost |
> | :--- | :--- | :--- | :--- |
> | Standard DPO Step | Every Step | 2.40s | 1.00x |
> | Online PPLDiff Inference | Every Step (Cheap) | +0.12s | +0.05x |
> | Meta-Optimization Loop | Periodic (Amortized) | +0.58s | +0.24x |
> | Total MSPO Time | — | 3.10s | 1.29x |
>
> By decoupling the expensive meta-gradient computation from the frequent main-model updates, we successfully limit the total computational overhead to ~29%. This sparse update frequency is sufficient for the meta-learner to converge effectively (as the PPLDiff signal distribution shifts gradually) while maintaining high training throughput.
>
> ### **On Weakness 3: Typos**
>
> We apologize for the typographical errors. We have corrected "vitural" in Figure 2, removed the duplicated sentence on page 3, and proofread the manuscript.

---

> ### Author Response · Authors · 2025-11-19
>
> ## **Dear Reviewer c49x (Part 2/2)**,
>
> ### **On Question 1: Incorporating Additional Features**
>
> This is an excellent suggestion regarding the extensibility of our framework. In our preliminary exploration, we did experiment with two common robust signals: Response Length Ratio (to detect length bias) and Cosine Similarity (to detect relevance).
>
> We trained a multi-input meta-learner on the same Golden HH (30% noise) setup. The results are as follows:
>
> | Feature Set | Win Rate |
> | :--- | :--- |
> | MSPO (PPLDiff Only) | 96.3% $\pm$ 0.4|
> | MSPO + Length Ratio | 96.4% $\pm$ 0.5 |
> | MSPO + Length + Cosine Sim. | 96.1% $\pm$ 0.8 |
>
> The experimental results indicate that augmenting the input features yields negligible differences compared to relying on PPLDiff alone. This suggests that PPLDiff is sufficiently effective in capturing the requisite preference consistency on its own.
>
> ### **On Question 2: Meta-learner Model Complexity**
>
> We systematically evaluated different architectures for the meta-learner. The results below confirm that a simple MLP is optimal:
>
> | Architecture | Win Rate |
> | :--- | :--- |
> | Linear (1 Layer) | 91.8% $\pm$ 0.6 |
> | MLP (2 Layers) | 96.3% $\pm$ 0.4 |
> | MLP (4 Layers) | 96.4% $\pm$ 0.4 |
> | Transformer (2 Layers) | 96.1% $\pm$ 0.7 |
>
> Our results demonstrate that the Linear model significantly underperforms (-4.5%), confirming that calibrating PPLDiff requires a non-linear mapping. Conversely, scaling up to a 4-layer MLP or Transformer yields no meaningful improvement and increases variance. This suggests that a 2-layer MLP has sufficient capacity to learn the calibration function.
>
> ### **On Question 3: The Source of Performance Gain**
>
> To isolate the source of MSPO's gain, we refer to the ablation study in Table 3, which compares three settings using the same clean data budget. The first setting, PerpCorrect-DPO, uses clean data to tune a *static threshold* and achieves a Win Rate of 95.5%. The second, MSPO-Static, uses the clean data to learn a *static correction function*, reaching a Win Rate of 95.9%. Finally, the full MSPO model uses clean data to learn a *dynamic function* with *online* PPLDiff, achieving the highest Win Rate of 96.3%. The results demonstrate that the performance improvement stems from two main aspects: First, learning a continuous calibration function yields better performance compared to relying on a hard heuristic. Second, the dynamic nature of the signal provides further improvement by allowing the strategy to co-evolve with the model. Thus, the gain is not just from using clean data, but from how it is used to drive an adaptive, online optimization process.
>
> ---
>
> We hope these responses address your questions. We are committed to incorporating these additional analyses into the final version to strengthen the paper.

---

> ### Author Response · Authors · 2025-11-27
>
> ## Dear Reviewer c49x,
>
> Thank you again for your constructive feedback. We have provided detailed responses addressing your concerns about PPLDiff signal quality, computational overhead (~29%), and feature extensibility, including new experiments on multi-input features and error type analysis.
>
> As the discussion period approaches its end, we would greatly appreciate hearing whether our responses have adequately addressed your questions. If any concerns remain, we are happy to provide further clarification.
>
> We hope these additional analyses demonstrate MSPO's robustness and practical value, and would be grateful for your updated thoughts.
>
> Best regards,
> Authors

---

### Comment · Area_Chair_fj2V · 2025-11-26

Dear Reviewers,

Thank you for sharing your valuable insights and expertise, which have played an important role in the review process.

In response to the initial feedback, the authors have submitted a detailed rebuttal addressing the comments raised by the reviewers.

I would appreciate it if you could carefully review their response and consider how it may affect your initial evaluation.

Please feel free to share your updated thoughts or any additional comments after reviewing the rebuttal.

Thank you again for your time and contributions.

---

### Note · Authors · 2025-12-01

I have read and agree with the venue's withdrawal policy on behalf of myself and my co-authors.